# Protein Representation Learning via Knowledge Enhanced Primary Structure Modeling

**Hong-Yu Zhou**[1*]  **Yunxiang Fu**[1,2*]  **Zhicheng Zhang**[3]  **Cheng Bian**[4]  **Yizhou Yu**[1†]

[1]Department of Computer Science, The University of Hong Kong
[2]Xiaohe Healthcare, ByteDance [3]JancsiTech [4]OPPO HealthLab
`whuzhouhongyu@gmail.com, yunxiang@connect.hku.hk,`
`zhangzhicheng13@mails.ucas.edu.cn, bian19920125@gmail.com,`
`yizhouy@acm.org`

## Abstract

Protein representation learning has primarily benefited from the remarkable development of language models (LMs). Accordingly, pre-trained protein models also suffer from a problem in LMs: a lack of factual knowledge. The recent solution models the relationships between protein and associated knowledge terms as the knowledge encoding objective. However, it fails to explore the relationships at a more granular level, i.e., the token level. To mitigate this, we propose Knowledge-exploited Auto-encoder for Protein (KeAP), which performs token-level knowledge graph exploration for protein representation learning. In practice, non-masked amino acids iteratively query the associated knowledge tokens to extract and integrate helpful information for restoring masked amino acids via attention. We show that KeAP can consistently outperform the previous counterpart on 9 representative downstream applications, sometimes surpassing it by large margins. These results suggest that KeAP provides an alternative yet effective way to perform knowledge enhanced protein representation learning. Code and models are available at `https://github.com/RL4M/KeAP`.

## 1 Introduction

The unprecedented success of AlphaFold (Jumper et al., 2021; Senior et al., 2020) has sparked the public's interest in artificial intelligence-based protein science, which in turn promotes scientists to develop more powerful deep neural networks for protein. At present, a major challenge faced by researchers is how to learn generalized representation from a vast amount of protein data. An analogous problem also exists in natural language processing (NLP), while the recent development of big language models (Devlin et al., 2018; Brown et al., 2020) offers a viable solution: unsupervised pre-training with self-supervision. In practice, by viewing amino acids as language tokens, we can easily transfer existing unsupervised pre-training techniques from NLP to protein, and the effectiveness of these techniques has been verified in protein representation learning Rao et al. (2019); Alley et al. (2019); Elnaggar et al. (2021); Unsal et al. (2022).

However, as pointed out by (Peters et al., 2019; Zhang et al., 2019; Sun et al., 2020; Wang et al., 2021), pre-trained language models often suffer from a lack of factual knowledge. To alleviate similar problems appearing in protein models, Zhang et al. (2022) proposed OntoProtein that explicitly injects factual biological knowledge into the pre-trained model, leading to observable improvements on several downstream protein analysis tasks, such as amino acid contact prediction and protein-protein interaction identification.

In practice, OntoProtein leverages the masked language modeling (MLM) (Devlin et al., 2018) and TransE (Bordes et al., 2013) objectives to perform structure and knowledge encoding, respectively. Specifically, the TransE objective is applied to triplets from knowledge graphs, where each triplet can be formalized as (*Protein*, *Relation*, *Attribute*). The relation and attribute terms described using natural language are from the gene ontologies (Ashburner et al., 2000) associated with protein.

---

[*]First two authors contributed equally. Yunxiang's work was done at ByteDance.
[†]Corresponding author.

However, OntoProtein only models the relationships on top of the contextual representations of protein (averaging amino acid representations) and textual knowledge (averaging word representations), preventing it from exploring knowledge graphs at a more granular level, i.e., the token level.

We propose KeAP (**K**nowledge-**e**xploited **A**uto-encoder for **P**rotein) to perform knowledge enhanced protein representation learning. To address the granularity issue of OntoProtein, KeAP performs token-level protein-knowledge exploration using the cross-attention mechanism. Specifically, each amino acid iteratively queries each word from relation and attribute terms to extract useful, relevant information using QKV Attention (Vaswani et al., 2017). The extracted information is then integrated into the protein representation via residual learning (He et al., 2016). The training process is guided only by the MLM objective, while OntoProtein uses contrastive learning and masked modeling simultaneously. Moreover, we propose to explore the knowledge in a cascaded manner by first extracting information from relation terms and then from attribute terms, which performs more effective knowledge encoding.

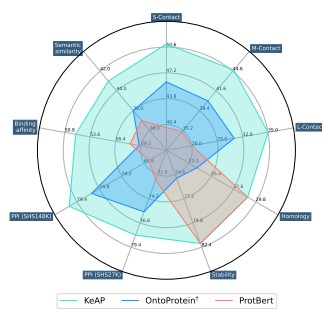

Figure 1: Transfer learning performance of ProtBert, OntoProtein, and our KeAP on downstream protein analysis tasks. S-, M-, and L-Contact stand for short-range, medium-range, and long-range contact prediction. PPI denotes the protein-protein interaction prediction. † means the model is trained with the full ProteinKG25.

KeAP has two advantages over OntoProtein (Zhang et al., 2022). First, KeAP explores knowledge graphs at a more granular level by applying cross-attention to sequences of amino acids and words from relation and attributes. Second, KeAP provides a neat solution for knowledge enhanced protein pre-training. The encoder-decoder architecture in KeAP can be trained using the MLM objective only (both contrastive loss and MLM are used in OntoProtein), making the whole framework easy to optimize and implement.

Experimental results verify the performance superiority of KeAP over OntoProtein. In Fig. 1, we fine-tune the pre-trained protein models on 9 downstream applications. We see that KeAP outperforms OntoProtein on all 9 tasks mostly by obvious margins, such as amino acid contact prediction and protein-protein interaction (PPI) identification. Compared to ProtBert, KeAP achieves better results on 8 tasks while performing comparably on protein stability prediction. In contrast, OntoProtein produces unsatisfactory results on homology, stability, and binding affinity prediction.

## 2  RELATED WORKS

### 2.1  REPRESENTATION LEARNING FOR PROTEIN

How to learn generalized protein representation has recently become a hot topic in protein science, inspired by the widespread use of representation learning in language models (Devlin et al., 2018; Radford et al., 2019; Yang et al., 2019; Sarzynska-Wawer et al., 2021). Bepler & Berger (2018) introduced a multi-task protein representation learning framework, which obtains supervision signals from protein-protein structural similarity and individual amino acid contact maps. Due to a plethora of uncharacterized protein data, self-supervised pre-training (Alley et al., 2019; Rao et al., 2019) was proposed to directly learn representation from chains of amino acids, where tremendous and significant efforts were made to improve the pre-training result by scaling up the size of the model and dataset (Elnaggar et al., 2021; Rives et al., 2021; Vig et al., 2020; Rao et al., 2020; Yang et al., 2022; Nijkamp et al., 2022; Ferruz et al., 2022; Chen et al., 2022). In contrast, protein-related factual knowledge, providing abundant descriptive information for protein, has been long ignored and largely unexploited. OntoProtein (Zhang et al., 2022) first showed that we can improve the performance of pre-trained models on downstream tasks by explicitly injecting the factual biological knowledge associated with protein sequences into pre-training.

In practice, OntoProtein proposed to reconstruct masked amino acids while minimizing the embedding distance between contextual representations of protein and associated knowledge terms. One potential pitfall of this operation is that it fails to explore the relationships between protein and

knowledge at a more granular level, i.e., the token level. In comparison, our KeAP overcomes this limitation by performing token-level protein-knowledge exploration via cross-attention modules.

## 2.2 KNOWLEDGE ENHANCED PRE-TRAINED LANGUAGE MODELS

Knowledge integration has been treated as a reliable way to improve modern language models (Zhang et al., 2019; Sun et al., 2020; Liu et al., 2019; Vulić et al., 2020; Petroni et al., 2019; Roberts et al., 2020; Wang et al., 2021; Yao et al., 2019; Liu et al., 2020; He et al., 2020; Qin et al., 2021; Madani et al., 2020; Chen et al., 2023; Zhou et al., 2022). Xie et al. (2016) proposed to perform end-to-end representation learning on triplets extracted from knowledge graphs, while Wang et al. (2021) further bridged the gap between knowledge graphs and pre-trained language models by treating entity descriptions as entity embeddings and jointly training the knowledge encoding (i.e., TransE (Bordes et al., 2013)) and MLM objectives.

Motivated by (Wang et al., 2021), OntoProtein (Zhang et al., 2022) used the MLM and TransE (Bordes et al., 2013) as two training objectives when learning protein representation on knowledge graphs. However, OntoProtein fails to explore the knowledge graphs at a more granular level, i.e., the token level. In contrast, KeAP performs token-level protein-knowledge exploration via the attention mechanism and provides a neat solution for knowledge enhanced protein pre-training by training an encoder-decoder architecture using only the MLM objective.

## 3 METHODOLOGIES

As shown in Fig. 2, for each triplet (*Protein*, *Relation*, *Attribute*) in the knowledge graphs, we apply random masking to the amino acid sequence while treating the relation and attribute terms as the associated factual knowledge. After feature extraction, representations of each triplet are sent to the protein decoder for reconstructing missing amino acids. The decoder model comprises $N$ stacked protein-knowledge exploration (PiK) blocks. In each block, the amino acid representation iteratively queries, extracts, and integrates helpful, relevant information from word representations of associated relation and attribute terms in a cascaded manner. The resulting representation is used to restore the masked amino acids, guided by the MLM objective. After pre-training, the protein encoder can be transferred to various downstream tasks.

### 3.1 PRELIMINARY: PROTEIN AND BIOLOGICAL KNOWLEDGE

KeAP is trained on a knowledge graph that consists of about five million triplets from ProteinKG25 (Zhang et al., 2022). Each triplet is in the format of (*Protein*, *Relation*, *Attribute*). *Protein* can be viewed as a sequence of amino acids, while both *Relation* and *Attribute* are factual knowledge terms (denoted as gene ontologies in Zhang et al. (2022)) described using natural language. Specifically, *Relation* and *Attribute* provide knowledge in biology that is associated with *Protein*, such as molecular function, biological process, and cellular components.

During the training stage, each protein is passed to the protein encoder, resulting in the protein representation $f_p^0 \in \mathbb{R}^{L_p \times D}$. The superscript 0 is the layer index. $L_p$ denotes the length of the amino acid sequence. $D$ stands for the feature dimension. In practice, the protein encoder has a BERT-like architecture (Devlin et al., 2018). Similarly, we forward the associated knowledge terms to the language encoder to obtain knowledge representations, i.e., $f_r \in \mathbb{R}^{L_r \times D}$ and $f_a \in \mathbb{R}^{L_a \times D}$. $L_r$ and $L_a$ denote the lengths of the relation and attribute terms, respectively. The reason for using two encoders is that we would like to extract domain-specific embeddings for protein and biological knowledge.

### 3.2 TOKEN-LEVEL PROTEIN-KNOWLEDGE EXPLORATION

KeAP uses a surrogate task to perform knowledge enhanced pre-training, i.e., exploring knowledge graphs for protein primary structure modeling. In this way, KeAP asks the protein representation to be aware of what knowledge is helpful to masked protein modeling. In practice, we treat each amino acid as a query, while the words from associated relation and attribute terms are attended to as keys and values in order. Taking the $i$-th layer as an example, the inputs to the protein decoder include

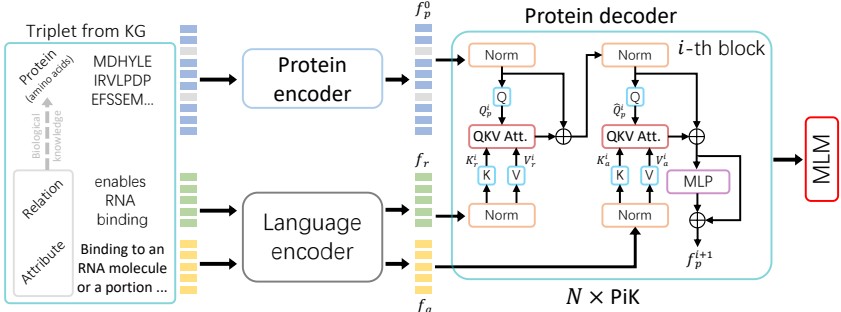

Figure 2: **Overview**. Given a triplet (*Protein*, *Relation*, *Attribute*) from the knowledge graph, KeAP randomly masks the input amino acid sequence and treats the relation and attribute terms as associated factual knowledge. Then, the triplet is passed to encoders and the output representations are regarded as the protein decoder's inputs. The decoder asks each amino acid representation to iteratively query words from the associated knowledge terms in a cascaded manner, extracting and integrating helpful, relevant information for restoring masked amino acids. **PiK** and **MLM** stand for the protein-knowledge exploration block and masked language modeling, respectively.

$f_p^i$, $f_r$, and $f_a$. The relation representation $f_r$ is firstly queried by $f_p^i$ as the key and value:

$$Q_p^i, \; K_r^i, \; V_r^i = \text{Norm}(f_p^i)W_Q^i, \; \text{Norm}(f_r)W_K^i, \; \text{Norm}(f_r)W_V^i, \tag{1}$$

where $W_Q^i$, $W_K^i$, and $W_V^i$ are learnable matrices. Norm stands for the layer normalization (Ba et al., 2016).

Then, QKV Attention (QKV-A) (Vaswani et al., 2017) is applied to $\{Q_p^i, \; K_r^i, \; V_r^i\}$, where representations of amino acids extract helpful, relevant knowledge from the latent word embeddings of the associated relation term. The obtained knowledge representation $s_p^i$ stores the helpful information for restoring missing amino acids. We then add up $s_p^i$ and $f_p^i$ to integrate knowledge, resulting in the representation $\hat{f}_p^i$.

$$\begin{aligned} s_p^i &= \text{QKV-A}(Q_p^i, \; K_r^i, \; V_r^i), \\ \hat{f}_p^i &= \text{Norm}(f_p^i) + s_p^i. \end{aligned} \tag{2}$$

Next, we use $\hat{f}_p^i$ to query the attribute term. The whole query, extraction, and integration process is similar to that of the relation term:

$$\begin{aligned} \hat{Q}_p^i, \; K_a^i, \; V_a^i &= \text{Norm}(\hat{f}_p^i)\hat{W}_Q^i, \; \text{Norm}(f_a)\hat{W}_K^i, \; \text{Norm}(f_a)\hat{W}_V^i, \\ \hat{s}_p^i &= \text{QKV-A}(\hat{Q}_p^i, \; K_a^i, \; V_a^i), \\ \bar{f}_p^i &= \text{Norm}(\hat{f}_p^i) + \hat{s}_p^i. \end{aligned} \tag{3}$$

The resulting representation $\bar{f}_p^i$ integrates the helpful, relevant biological knowledge that benefits the restoration of missing amino acids. We finally forward $\bar{f}_p^i$ through a residual multi-layer perceptron (MLP) to obtain the output representation of the $i$-th PiK block, which also serves as the input to the $i+1$-th block.

## 3.3 MASKED LANGUAGE MODELING OBJECTIVE

For each input protein, we randomly mask 20% amino acids in the sequence. Moreover, each masked amino acid has an 80% chance of being masked for prediction, a 10% chance of being replaced by a random amino acid, and a 10% chance of remaining unchanged. Suppose the number of masked amino acids is $M$ and $x_j$ denotes the $j$-th amino acid. The training objective $\mathcal{L}_{\text{MLM}}$ to be minimized is as follows:

$$\mathcal{L}_{\text{MLM}} = -\log \sum_{j=0}^{M-1} P\left(x_j \mid f_p^N; \; \Theta_E, \Theta_D\right). \tag{4}$$

$\Theta_E$ and $\Theta_D$ denote the parameters of the protein encoder and decoder, respectively. We initialize the language encoder using PubMedBERT (Gu et al., 2021) and do not update it in the training phase.

## 4 EXPERIMENTS AND ANALYSES

In this section, we extensively evaluate the generalization ability of the learned protein representation by fine-tuning the pre-trained model on a wide range of downstream applications, including amino acid contact prediction, protein homology detection, protein stability prediction, protein-protein interaction identification, protein-protein binding affinity prediction, and semantic similarity inference. Besides, we also provide ablations and failure analyses to facilitate the understanding of KeAP. Unless otherwise specified, we follow the pre-training and fine-tuning protocols used by OntoProtein (refer to the appendix for more details), such as training strategies and dataset split. The pre-trained models of ProtBert (Elnaggar et al., 2021), OntoProtein (Zhang et al., 2022), and our KeAP share the same number of network parameters. Average results are reported over three independent training runs.

### 4.1 DATASET FOR PRE-TRAINING

ProteinKG25 (Zhang et al., 2022) provides a knowledge graph that consists of approximately five million triplets, with nearly 600k protein, 50k attribute terms, and 31 relation terms included. The attribute and relation terms described using natural language are extracted from Gene Ontology [1] which is the world's largest source of information on the functions of genes and gene products (e.g., protein).

It is worth noting that we pre-train KeAP and report its experimental results in two different settings where the pre-training datasets differ. In the first setting, we remove 22,978 amino acid sequences that appear in downstream tasks. Accordingly, 396,993 triplets are removed from ProteinKG25, and the size of the new dataset is 91.87% of the original. In the second setting (denoted with [†]), we use the same pre-training data as in OntoProtein (Zhang et al., 2022).

### 4.2 AMINO ACID CONTACT PREDICTION

| Methods | $6 \leq seq < 12$ | | | $12 \leq seq < 24$ | | | $24 \leq seq$ | | |
|---|---|---|---|---|---|---|---|---|---|
| | P@L | P@L/2 | P@L/5 | P@L | P@L/2 | P@L/5 | P@L | P@L/2 | P@L/5 |
| LSTM | 0.26 | 0.36 | 0.49 | 0.20 | 0.26 | 0.34 | 0.20 | 0.23 | 0.27 |
| ResNet | 0.25 | 0.34 | 0.46 | 0.28 | 0.25 | 0.35 | 0.10 | 0.13 | 0.17 |
| Transformer | 0.28 | 0.35 | 0.46 | 0.19 | 0.25 | 0.33 | 0.17 | 0.20 | 0.24 |
| ProtBert | 0.30 | 0.40 | 0.52 | 0.27 | 0.35 | 0.47 | 0.20 | 0.26 | 0.34 |
| ESM-1b | 0.38 | 0.48 | 0.62 | 0.33 | 0.43 | **0.56** | 0.26 | 0.34 | **0.45** |
| KeAP | **0.41** | **0.51** | **0.63** | **0.36** | **0.45** | 0.54 | **0.28** | **0.35** | 0.43 |
| OntoProtein[†] | 0.37 | 0.46 | 0.57 | 0.32 | 0.40 | 0.50 | 0.24 | 0.31 | 0.39 |
| KeAP[†] | **0.41** | **0.52** | **0.62** | **0.36** | **0.46** | **0.57** | **0.29** | **0.37** | **0.46** |

Table 1: Comparisons on amino acid contact prediction. **seq** indicates the distance (i.e., the number of amino acids) between two selected amino acids. **P@L**, **P@L/2**, **P@L/5** denote the precision scores calculated upon top L (i.e., L most likely contacts), top L/2, and top L/5 predictions, respectively. In each setting, the best results are bolded. Models with † are pre-trained using the full ProteinKG25.

**Overview.** Given an input protein molecule that comprises a chain of amino acids, the protein model is asked to predict whether any two amino acids (from the same sequence) are in contact or not. To achieve this goal, our model outputs a probability contact matrix for each input protein, where the row and column numbers correspond to the indices of two amino acids. We performed experiments on the dataset collected and organized by (AlQuraishi, 2019; Rao et al., 2019). The evaluation metric is precision.

---

[1]http://geneontology.org/

**Baselines.** Following (Zhang et al., 2022), we included six protein analysis models as baselines. Specifically, we used variants of LSTM (Hochreiter & Schmidhuber, 1997), ResNet (He et al., 2016), and Transformer (Vaswani et al., 2017) proposed by the TAPE benchmark (Rao et al., 2019). ProtBert (Elnaggar et al., 2021) is a 30-layer BERT-like model pre-trained on UniRef100 (Suzek et al., 2007; 2015). ESM-1b is a 33-layer transformer pre-trained on UR50/S, which uses the UniRef50 (Suzek et al., 2007; 2015) representative sequences. OntoProtein Zhang et al. (2022) is the most recent knowledge-based pre-training methodology.

**Results.** Table 1 presents the experimental results of amino acid contact prediction. In the first setting, we see that ProtBert and ESM-1b are two most competitive baselines. Specifically, KeAP outperforms ProtBert by large margins in short- ($6 \leq seq < 12$), medium- ($12 \leq seq < 24$), and long-range ($seq \geq 24$) contact predictions. Compared to ESM-1b, KeAP is more advantageous in short-range prediction while achieving competitive results in medium- and long-range prediction tasks. In the second setting, we see that KeAP consistently surpasses OntoProtein regardless of the the distance between two amino acids. Particularly noteworthy is the fact that KeAP surpasses OntoProtein by large margins. Considering that OntoProtein also performs knowledge encoding, the clear performance advantages (6%) over OntoProtein demonstrate that KeAP provides a more competitive choice for knowledge enhanced protein representation learning. We believe the performance gains brought by KeAP can be attributed to the proposed token-level knowledge graph exploration methodology. This enables the pre-trained model to understand the knowledge context better, producing a better-contextualized protein representation for contact prediction than directly applying meaning pooling.

## 4.3 HOMOLOGY DETECTION AND STABILITY PREDICTION

**Overview of homology detection.** Predicting the remote homology of protein can be formalized as a molecule-level classification task. Given a protein molecule as the input, we ask the homology detection model to predict the right protein fold type. In our case, there are 1,195 different types of protein folds, making it a quite challenging task. The data are from Hou et al. (2018) and we report average accuracy on the fold-level heldout set.

**Overview of stability prediction.** In this regression task, we aim to predict the intrinsic stability of the protein molecule, which measures the protein's ability to maintain its fold under extreme conditions. In practice, high-accuracy stability prediction can benefit drug discovery, making it easier to construct stable protein molecules. We evaluate the model performance by calculating Spearman's rank correlation scores on the whole test set Rocklin et al. (2017).

| Methods | Homology | Stability |
|---|---|---|
| LSTM | 0.26 | 0.69 |
| ResNet | 0.17 | 0.73 |
| Transformer | 0.21 | 0.73 |
| ProtBert | **0.29** | **0.82** |
| ProteinBert | 0.22 | 0.76 |
| ESM-1b | 0.11 | 0.77 |
| KeAP | **0.29** | **0.82** |
| OntoProtein† | 0.24 | 0.75 |
| KeAP† | **0.30** | **0.82** |

Table 2: Comparisons on protein homology detection and stability prediction. In each setting, the best results are bolded. Models with † are pre-trained using the full ProteinKG25.

**Baselines.** Besides six approaches presented in Table 1, we add one more baseline ProteinBert (Brandes et al., 2022), which pre-trains the protein model to restore missing amino acids and associated attribute annotations simultaneously.

**Results.** From Table 2, we see that the knowledge-based pre-training methodologies, i.e., ProteinBert and OntoProtein, fail to display favorable results on both tasks. Zhang et al. (2022) claimed that the failure is due to the lack of sequence-level objectives in pre-training. In contrast, our KeAP performs on par with ProtBert on homology detection and protein stability prediction. We believe the success of KeAP can be partly attributed to the token-level knowledge graph exploration.

## 4.4 PROTEIN-PROTEIN INTERACTION IDENTIFICATION

**Overview.** Protein-protein interactions (PPI) refer to the physical contacts between two or more amino acid sequences. In this paper, we only study the two-protein cases, where a pair of protein molecules serve as the inputs. The goal is to predict the interaction type(s) of each protein pair. There

| Methods | SHS27K | | | SHS148K | | | STRING | | |
|---|---|---|---|---|---|---|---|---|---|
| | BFS | DFS | Avg | BFS | DFS | Avg | BFS | DFS | Avg |
| DNN-PPI | 48.09 | 54.34 | 51.22 | 57.40 | 58.42 | 57.91 | 53.05 | 64.94 | 59.00 |
| DPPI | 41.43 | 46.12 | 43.77 | 52.12 | 52.03 | 52.08 | 56.68 | 66.82 | 61.75 |
| PIPR | 44.48 | 57.80 | 51.14 | 61.83 | 63.98 | 62.91 | 55.65 | 67.45 | 61.55 |
| GNN-PPI | 63.81 | 74.72 | 69.27 | 71.37 | 82.67 | 77.02 | 78.37 | **91.07** | 84.72 |
| ProtBert | 70.94 | 73.36 | 72.15 | 70.32 | 78.86 | 74.59 | 67.61 | 87.44 | 77.53 |
| ESM-1b | 74.92 | **78.83** | 76.88 | **77.49** | 82.13 | 79.31 | 78.54 | 88.59 | 83.57 |
| KeAP | **78.58** | 77.54 | **78.06** | 77.22 | **84.74** | **80.98** | **81.44** | 89.77 | **85.61** |
| OntoProtein[†] | 72.26 | 78.89 | 75.58 | 75.23 | 77.52 | 76.38 | 76.71 | **91.45** | 84.08 |
| KeAP[†] | **79.17** | **79.77** | **79.47** | **75.67** | **83.04** | **79.36** | **80.78** | 89.02 | **84.90** |

Table 3: Comparisons on PPI identification. Experiments were performed on three datasets, whose F1 scores are presented. The best results in each setting are bolded. Models with † are pre-trained using the full ProteinKG25.

are 7 types included in experiments, which are reaction, binding, post-translational modifications, activation, inhibition, catalysis, and expression. The problem of PPI prediction can be formalized as a multi-label classification problem. We perform experiments on SHS27K (Chen et al., 2019), SHS148K (Chen et al., 2019), and STRING (Lv et al., 2021). SHS27K and SHS148K can be regarded as two subsets of STRING, where protein with fewer than 50 amino acids or $\geq 40\%$ sequence identity is excluded. Breadth-First Search (BFS) and Depth-First Search (DFS) are used to generate test sets from the aforementioned three datasets. F1 score is used as the default evaluation metric.

**Baselines.** Following Zhang et al. (2022), we introduce DPPI (Hashemifar et al., 2018), DNN-PPI (Li et al., 2018), PIPR (Chen et al., 2019), and GNN-PPI (Lv et al., 2021) as 4 more baselines in addition to ProtBert, ESM-1b, and OntoProtein.

**Results.** Experimental results are displayed in Table 3. In the first setting, we see that ESM-1b is the best performing baseline. Nonetheless, our KeAP still achieves the best average performance, outperforming ESM-1b on all three datasets by 1% at least. In the second setting, KeAP outperforms OntoProtein by about 4%, 3%, and 1% on SHS27K, SHS148K, and STRING, respectively. The trend of declining performance can be attributed to the increasing amount of fine-tuning data (from SHS27K to STRING) that reduces the impact of pre-training. Specifically, the advantages of KeAP are quite obvious on SHS27K which has the least number of protein, indicating the effectiveness of the protein representation from KeAP with limited fine-tuning data. As the amount of training data increases (from SHS27K to STRING), ProtBert and OntoProtein gradually display inferior performance, compared to GNN-PPI. In contrast, our KeAP still performs competitively and surpasses GNN-PPI by an obvious margin on BFS.

## 4.5 PROTEIN-PROTEIN BINDING AFFINITY ESTIMATION

**Overview.** In this task, we aim to evaluate the ability of the protein representation to estimate the change of the binding affinity due to mutations of protein. In practice, each pair of protein is mapped to a real value (thus this is a regression task), indicating the binding affinity change. We follow (Unsal et al., 2022) to apply bayesian ridge regression to the result of the element-wise multiplication of representation extracted from pre-trained protein models for predicting the binding affinity. We used the SKEMPI dataset from (Moal & Fernández-Recio, 2012) and report the mean square error of 10-fold cross-validation.

| Methods | Affinity ($\downarrow$) |
|---|---|
| PIPR | 0.63 |
| ProtBert | 0.58 |
| ESM-1b | **0.50** |
| KeAP | 0.52 |
| OntoProtein[†] | 0.59 |
| KeAP[†] | 0.51 |

Table 4: Comparisons on protein-protein binding affinity prediction. The best result in each setting is bolded. $\downarrow$ means the lower the better. Models with † are pre-trained using the full ProteinKG25.

**Baselines.** We include PIPR, ProtBert, ESM-1b, and OntoProtein as comparative baselines.

**Results.** Table 4 presents the experimental results on the binding affinity estimation task. We see that KeAP outperforms PIPR, ProtBert, and ESM-1b by substantial margins. Considering the region-level structural feature plays a vital role in this task (Unsal et al., 2022), we believe the obvious performance advantage of KeAP again verifies the effectiveness of the proposed token-level knowledge graph exploration methodology. Nonetheless, there is still a performance gap between our KeAP and ESM-1b, which may be attributed to the difference in network architecture.

## 4.6 Semantic Similarity Inference

**Overview.** In this task, given two interacting protein molecules and their associated attribute terms, we first calculate the Manhattan Similarity[2] between their representations. Then, we calculate the Lin Similarity between their associated attribute terms following instructions from Unsal et al. (2022). Finally, Spearman's rank correlation is calculated between the Manhattan Similarity scores and Lin Similarity scores, where the Lin Similarity scores are treated as ground truths, and the Manhattan Similarity scores are regarded as predictions. Specifically, we divide protein attributes into three groups: molecular function (MF), biological process (BP), and cellular component (CC), and report the correlation scores for each group in Table 5.

| Methods | MF | BP | CC | Avg |
|---|---|---|---|---|
| MSA Transformer | 0.38 | 0.31 | 0.30 | 0.33 |
| ProtT5-XL | **0.57** | 0.21 | **0.40** | 0.39 |
| ProtBert | 0.41 | 0.35 | 0.36 | 0.37 |
| ESM-1b | 0.38 | **0.42** | 0.37 | 0.39 |
| KeAP | 0.42 | 0.41 | 0.39 | **0.41** |
| OntoProtein[†] | **0.41** | 0.36 | 0.36 | 0.38 |
| KeAP[†] | **0.41** | **0.41** | **0.40** | **0.41** |

Table 5: Comparisons on semantic similarity inference. The best results in each setting are bolded. Models with † are pre-trained using the full ProteinKG25.

**Baselines.** In addition to ProtBert and OntoProtein, we introduce three powerful pre-trained protein models for comparisons, which include MSA Transformer (Rao et al., 2021), ESM-1b (Rives et al., 2021), and ProtT5-XL (Elnaggar et al., 2021).

**Results.** Table 5 presents the similarity inference results. We see that KeAP achieves the best average result even compared to larger (with more parameters) protein models, such as ESM-1b and ProtT5-XL. Specifically, ProtT5-XL produces the best performance on MF and CC, while ESM-1b performs the best on BP. Compared to ESM-1b and ProtT5-XL, our KeAP gets 2nd place on MF and achieves the highest score on CC. These results demonstrate the potential of KeAP in outperforming big protein models, and it would be interesting if KeAP could be integrated into bigger models. Again, KeAP outperforms OntoProtein by substantial margins.

| | Contact | Homology | Stability | PPI | Affinity | Semantic Similarity |
|---|---|---|---|---|---|---|
| Performance changes | -1% | -1% | 0% | +1% | -1% | 0% |

Table 6: Performance changes when removing the proteins that appear in downstream tasks from the pre-training dataset (ProteinKG25). Average results are reported on contact prediction and PPI tasks.

| Ratios | Contact | Homology | PPI |
|---|---|---|---|
| 15% | 0.44 | 0.27 | 76.72 |
| 20% | 0.45 | 0.29 | 78.06 |
| 25% | 0.46 | 0.28 | 77.34 |

Table 7: Ablations of mask ratios. Medium-range P@L/2 results are reported for contact prediction.

| Strategies | Contact |
|---|---|
| KeAP | 0.45 |
| − Cascaded | 0.44 |
| − PiK | 0.38 |
| + Tri. Match | 0.45 |

Table 8: Investigation of knowledge exploitation strategies.

| Methods | SS-Q3 | SS-Q8 | Fluorescence |
|---|---|---|---|
| ProtBert | 0.81 | 0.67 | 0.67 |
| ESM-1b | - | 0.71 | 0.65 |
| KeAP | 0.82 | 0.67 | 0.65 |
| OntoProtein[†] | 0.82 | 0.68 | 0.67 |
| KeAP[†] | 0.82 | 0.68 | 0.67 |

Table 9: Failure case analysis. We report the results on three tasks from (Rao et al., 2019).

---

[2]Manhattan Similarity = 1 - Manhattan Distance (normalized).

## 5 ABLATION AND DISCUSSION

We present ablation study results in Tables 6, 7, and 8, where we investigate the impacts of using different pre-training datasets, different mask ratios, and different knowledge integration strategies, respectively. Table 9 presents the performance on three tasks from Rao et al. (2019). SS-Q3 and SS-Q8 are two secondary structure prediction tasks (Klausen et al., 2019; Cuff & Barton, 1999) with different numbers of local structures. Fluorescence is a regression task, where the model is asked to predict the log-fluorescence intensity of each protein.

### 5.1 ABLATION STUDY

Table 6 presents the performance drops when we remove the proteins that appear in downstream tasks from ProteinKG25. We see that the pre-trained model shows slightly worse results on three out of six downstream tasks: contact prediction, homology detection, and affinity prediction, while performing competitively on PPI, stability prediction, and semantic similarity inference. These comparisons imply that we may not have to worry too much about the consequences of data leakage. We will continue to investigate this problem in future work.

As shown in Table 7, the 20% mask ratio performs the best on two (homology and PPI) of the three downstream tasks, which is the primary reason that we choose 20% as the default mask ratio. It is interesting that a larger ratio (i.e., 25%) leads to better performance on the contact prediction task. We leave the exploration of larger mask ratios for future work.

In addition to the mask ratio, we also study the impacts of using different knowledge exploitation strategies. First of all, removing the cascaded exploitation strategy (presented as - Cascaded in Table 8) results in a 1-percent performance drop on contact prediction, implying exploring the factual knowledge in a cascaded manner is a more effective choice. Then, we remove the proposed protein-knowledge exploration block (denoted as - PiK in Table 8), which means KeAP is simplified to an auto-encoder trained with the MLM objective. We find that this activity leads to an 7-percent performance drop on the contact prediction task, reflecting the necessity of incorporating knowledge into protein pre-training. Besides, we add a Triplet Matching training objective (appeared as - Tri. Match in Table 8) to KeAP, where we randomly replace the associated attribute term with a different one and train the model to tell whether the input triplet is matched or not. The idea is similar to that of the knowledge-aware contrastive learning proposed by OntoProtein (Zhang et al., 2022), which is learning the knowledge-aware protein representation. From Table 8, we see that adding the matching objective does not bring performance improvements to KeAP, indicating that our proposed exploration strategy for knowledge graphs may already master the information introduced by the Triplet Matching objective.

### 5.2 FAILURE CASE ANALYSIS

We report the experimental results on three tasks from Rao et al. (2019), where KeAP performs on par with or worse than ProtBert, ESM-1b, or OntoProtein. Specifically, in SS-Q3 and SS-Q8, the model is asked to predict the secondary structure of each amino acid, which heavily relies on the local information contained in the protein representation. We think the non-significant performance of KeAP is due to the lack of the incorporation of local details when performing the knowledge encoding. Similarly, on the Fluorescence task, KeAP also fails to achieve observable progress when asked to distinguish very similar protein molecules. Considering the same issues also exist in ProtBert and OntoProtein, we believe it is necessary to pay more attention to how to improve the performance on local prediction tasks by integrating more local information during the pre-training stage. We will continue to explore this issue in the future.

## 6 CONCLUSION AND FUTURE WORK

KeAP performs token-level knowledge graph exploration using cross-modal attention, providing a neat solution for knowledge enhanced protein pre-training. KeAP outperforms the previous knowledge enhanced counterpart on 9 downstream applications, sometimes by substantial margins, demonstrating the performance superiority of KeAP. In the future, we will investigate how to deploy KeAP on specific applications where the factual knowledge makes a greater impact.

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

# A APPENDIX

## A.1 HYPER-PARAMETERS FOR FINE-TUNING

The hyper-parameters for fine-tuning are provided in Table 10. Specifically, we follow the hyper-parameter settings in GNN-PPI (Lv et al., 2021) for PPI prediction. For protein binding affinity prediction and semantic similarity inference, we follow the fine-tuning configurations in PROBE (Unsal et al., 2022).

| Task | Epoch | Batch size | Warmup ratio | Learning rate | Freeze Bert | Optimizer |
|------|-------|-----------|--------------|---------------|-------------|-----------|
| Contact | 5 | 8 | 0.08 | 3e-5 | false | AdamW |
| Homology | 10 | 32 | 0.08 | 4e-5 | false | AdamW |
| Stability | 5 | 64 | 0.08 | 1e-5 | false | AdamW |
| SS-Q3 | 5 | 32 | 0.08 | 3e-5 | false | AdamW |
| SS-Q8 | 5 | 32 | 0.08 | 3e-5 | false | AdamW |
| Fluorescence | 15 | 64 | 0.10 | 1e-3 | true | Adam |

Table 10: Hyper-parameters for fine-tuning.

## A.2 PROBABILITY CONTACT MAPS

In Fig. 3, we present two randomly picked samples for visual analysis, where KeAP is able to detect contacts missed by ProtBert and OntoProtein with high confidence.

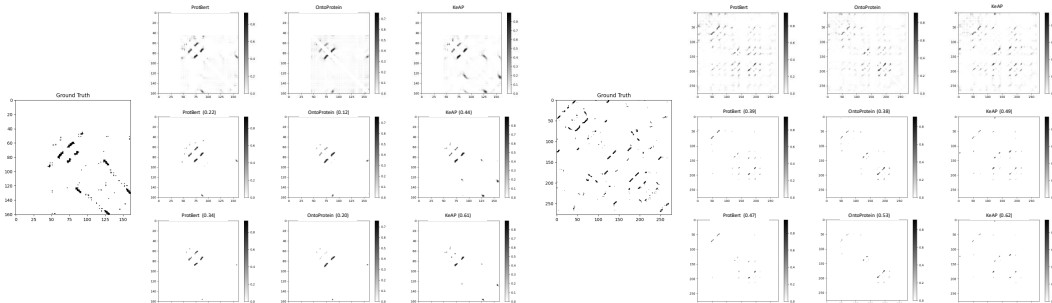

Figure 3: Ground truths and predicted probability contact maps. We compare the predictions of ProtBert and OntoProtein with ours, where the 1st, 2nd, and 3rd rows include the all, top L, and top L/2 predictions, respectively. For top L and L/2 predictions, the precision scores are reported. More examples are in the appendix.

