# OpenReview forum: "Protein Representation Learning via Knowledge Enhanced Primary Structure Reasoning"
_ICLR.cc/2023/Conference — ICLR 2023 poster_

### Official Review · Reviewer_cqj2 · 2022-10-14

**Confidence:** 4
**Correctness:** 3
**Technical Novelty And Significance:** 3
**Empirical Novelty And Significance:** 2
**Recommendation:** 6

**Clarity, Quality, Novelty And Reproducibility:**

Quality:

In general, this paper is well-written. The motivation and derivation of proposed techniques are clear to the audience.


Novelty:

The idea of enhancing protein language models with biomedical texts is novel and worth more explorations in the future.


Reproducibility:

Authors submitted source codes for reproducing the results. The codes look well-organized according to my brief check.


**Strength And Weaknesses:**

Strength:
1. It is a novel idea to enhance protein representations with associated biomedical texts (e.g., the relevant molecular functions and biological processes). The proposed cross-attention fusion scheme is technically sound to achieve this goal.
2. KeAP achieves some obvious performance improvements over the previous SOTA OntoProtein on contact prediction and PPI prediction tasks.


Weakness:
1. KeAP does not fully utilize the biomedical texts associated with a protein. For each forward pass, KeAP adopts a single (protein, relation, attribute) triplet and enhance the protein representation with only the relation and attribute texts in this triplet. However, in the biological KG used by authors, each protein can appear in multiple triplets, where these triplets describe orthogonal properties of the protein. Therefore, it is more promising to combine different text descriptions associated with a protein and inject relevant textual information across the combined text. I am curious and willing to see the performance of KeAP under such an improved input format.
2. Some important baselines are missed in performance comparison. Authors are suggested to include the results of the SOTA protein language models, ESM-1b and ProtT5, into Tables 1, 2 and 3.


**Summary Of The Paper:**

This paper proposes a novel protein sequence pre-training method, Knowledge-exploited Auto-encoder for Proteins (KeAP). Instead of injecting biomedical knowledge by directly applying knowledge embedding constraints as OntoProtein, KeAP employ a cross-attention mechanism to attentively inject relevant relation and attribute representations into protein representations. Upon such enhanced protein representations, masked residue type prediction loss is applied for representation learning. On contact prediction, homology detection, stability prediction and PPI prediction benchmarks, authors demonstrate the superiority of KeAP over OntoProtein and ProtBert (the initial protein sequence encoder of KeAP).

**Summary Of The Review:**

In summary, I am convinced by the motivation of learning better protein representations guided by biomedical texts and the cross-attention technique to achieve this goal. However, I have concerns on the input data organization during pre-training and the completeness of performance comparison. Therefore, I think the current manuscript is on the border and expect authors' efforts during rebuttal.

---

> ### Author Response · Authors · 2022-11-17
> **Response to Reviewer cqj2**
>
> We sincerely appreciate your reception of our technical novelty, performance improvements, and paper writing. We attach a point-by-point response below to address your concerns. If you have further questions or suggestions, please do let us know.
>
> ----
> ### Comment-1:
> > KeAP does not fully utilize the biomedical texts associated with a protein. For each forward pass, KeAP adopts a single (protein, relation, attribute) triplet and enhance the protein representation with only the relation and attribute texts in this triplet. However, in the biological KG used by authors, each protein can appear in multiple triplets, where these triplets describe orthogonal properties of the protein. Therefore, it is more promising to combine different text descriptions associated with a protein and inject relevant textual information across the combined text. I am curious and willing to see the performance of KeAP under such an improved input format.
>
> ### Response-1:
>
> Thanks for your insightful comments and constructive advice. As you mentioned, in the knowledge graph, each protein is often associated with more than one triplet (i.e., the same protein with varying relation and attribute terms). **Given a specific protein, KeAP does adopt all associated triplets for pre-training, although these triplets are passed to KeAP in different orders.** We will continue to explore this direction in future work.
>
> ----
> ### Comment-2:
> > Some important baselines are missed in performance comparison. Authors are suggested to include the results of the SOTA protein language models, ESM-1b and ProtT5, into Tables 1, 2 and 3.
>
> ### Response-2:
>
> We appreciate your kind suggestions. We mainly compare our method against ProtBert because they share the same network architecture, which can demonstrate the performance gains brought by knowledge-based pre-training. We have tried our best to include the results of SOTA protein language models. However, due to the limited time and computational resources, we can only provide the results of ESM-1b in the updated manuscript. Compared to ESM-1b, our KeAP still achieves noticeable improvements in a range of downstream tasks, including contact prediction, homology detection, stability prediction, protein-protein interaction identification, and semantic similarity inference.

---

> > ### Comment · Reviewer_cqj2 · 2022-11-26
> > **Feedback to Author Response**
> >
> > Hi authors,
> >
> > Thanks for your efforts during the rebuttal phase. I appreciate the new ESM-1b baseline results in the revision, which forms a more comprehensive comparison on multiple benchmarks.
> >
> > I notice the concerns about data leakage. However, considering the downstream tasks (contact predcition, PPI prediction, remote homology detection, etc.) are less correlated with the pre-training BioKG based on Gene Ontology, I think this leakage does not provide a short cut for downstream tasks. Therefore, I remain the borderline accept rating.

---

> > > ### Author Response · Authors · 2022-11-26
> > > **Reply to Reviewer cqj2**
> > >
> > > Dear Reviewer cqj2,
> > >
> > > We sincerely appreciate your reply, which does encourage us a lot! We will continue to investigate knowledge enhanced protein pre-training in future work.
> > >
> > > Again, thanks for your reply.
> > >
> > > Best,
> > >
> > > Paper1625 Authors

---

> ### Author Response · Authors · 2022-11-18
> **Follow up**
>
> Dear Reviewer cqj2,
>
> Thank you for your time and effort in reading our response! We hope our response has addressed your concerns. If you still feel unclear or concerned, please let us know, and we will be more than glad to clarify and discuss any further concerns. If you feel your concerns have been addressed, please kindly consider if it is possible to update your score.
>
> Thank you!
>
> Paper1625 Authors

---

### Official Review · Reviewer_mLUJ · 2022-10-19

**Confidence:** 5
**Correctness:** 2
**Technical Novelty And Significance:** 2
**Empirical Novelty And Significance:** 3
**Recommendation:** 6

**Clarity, Quality, Novelty And Reproducibility:**

Well written paper and original work.
There is an issue with the description of implementation details.


**Strength And Weaknesses:**

Strengths
This a well-written paper.
Significant experimental results compared to OntoProtein.
A good application paper.

Weaknesses

Additional comparisons to state-of-the-art language models are needed:
Recent language models for proteins such as ESM-1B, ESM-2, and MSA-Transformer demonstrate better results on TAPE benchmark datasets. I see you compare your approach with these models on only similarity prediction tasks. Why don't you compare these methods to the other benchmark datasets like contact prediction, homology, stability, protein-protein interaction, and affinity binding?

A potential flaw in experiments:
Potential leakage happens if proteins in the test sets of the benchmark datasets are not removed from the training, do you check the overlapping between test sets and the ProteinKG25. If that is not the case, the reported results might be over-optimistic.

Related to semantic gaps:
The authors claim that semantic gaps are the main issues of OntoProtein but it is very hard to see any evidence demonstrated in the paper to support that claim. Since this is the main claim against OntoProtein, this needs to be supported by a piece of strong evidence with careful analysis.

Reproducity:
I checked both the paper and the appendix version, this information is needed to clarify the implementation details:
+ what is the representation of proteins used in the encoder? I understand it is a bert-like encoder but whether you use pretrained models or train everything from scratch?
+ potential leakage happens if proteins in the test sets of the benchmark datasets are not removed from the training, do you check the overlapping between test sets and the ProteinKG25 ?



**Summary Of The Paper:**

Summary of the paper
OntoProtein is prior art that integrates additional information from Gene Ontology into protein presentation. OntoProtein chooses the MLM and TransE as two training objectives when learning protein representation on the Gene Ontology knowledge graphs.
The authors claim that the TransE objective has an issue because, in Gene Ontology, a triple usually concerns (Protein, Relation in natural language, and Property in natural language). While the initial embedding space of proteins and texts differs,  "OntoProtein did not take into account the domain gap between protein and natural language, and this semantic gap may make the TransE objective a suboptimal choice ".
Therefore, the authors proposed a new learning objective where only masked language model loss is considered during the training process. Specifically, random amino acids are masked as usual. Initial embeddings of masked protein sequences and texts (relations and properties) are obtained using pretrained models for proteins and texts respectively.
These initial embeddings are used by a decoder where attention layers (QKV Vaswani et al.) are used to obtain relation/property attention, given the initial embedding of proteins as a query.  The main objective of the decoder is to learn attentive weights to knowledge graph triples associated with the protein such that it helps predict masked amino acids in the sequence.
The authors claimed that this solution learned to incorporate knowledge implicitly into the protein representation and solved the semantic gap issues. The authors demonstrated the significance of the results compared to OntoProtein in 9 different downstream tasks.



**Summary Of The Review:**

Summary of the comments:
Although I see this as an interesting research work I am concerned about its evaluation methods with potential leakage. I am also concerned about its baseline methods to compare. More implementation details are needed to reproduce the results and stronger evidence to support the claim about "semantic gaps"

---

> ### Author Response · Authors · 2022-11-17
> **Response to Reviewer mLUJ (PART 1)**
>
> We sincerely appreciate your reception of the paper writing, performance improvements, and empirical novelty. Below we attach a point-by-point response to address your concerns. If you have further questions or suggestions, please do let us know.
>
> ----
> ### Comment-1:
>
> > Additional comparisons to state-of-the-art language models are needed.
>
> ### Response-1:
>
> Thanks for your kind advice. We mainly compare our method against ProtBert because they share the same network architecture, which can demonstrate the performance gains brought by knowledge-based pre-training.
>
> We agree that adding a recent language model for proteins would be helpful for comparison. However, due to the limited time and computational resources, we can only afford to do experiments on ESM-1B, whose results have been added to the revision. Compared to ESM-1b, our KeAP still achieves noticeable improvements in a range of downstream tasks, including contact prediction, homology detection, stability prediction, protein-protein interaction identification, and semantic similarity inference.
>
> ----
> ### Comment-2:
>
> > A potential flaw in experiments: Potential leakage happens if proteins in the test sets of the benchmark datasets are not removed from the training, do you check the overlapping between test sets and the ProteinKG25. If that is not the case, the reported results might be over-optimistic.
>
> ### Response-2:
> Thanks for your insightful comments. Previously, **we used the same pre-training data as used in OntoProtein [ICLR'22]. We also used the same data splits on contact prediction, homology detection, and stability prediction as used in OntoProtein.** According to your comments, we checked the overlap between the pre-training data (ProteinKG25) and downstream datasets. We found that there are 22,978 amino acid sequences that appear both in ProteinKG25 and all downstream datasets. Accordingly, 396,993 triplets are removed from ProteinKG25, and the size of the new dataset is 91.87% of the original.
>
> We report the experimental results in the updated manuscript based on the new pre-trained model trained with the filtered ProteinKG25. We also conduct ablation studies to investigate the impact of the leaked protein (please refer to section 5.1 for more details). We provide an excerpt here for your consideration:
>
> ||Contact|Homology|Stability|PPI|Affinity|Semantic Similarity|
> |----|----|----|----|----|----|----|
> |Performance changes|-1%|-1%|0%|+1|-1%|0%|
>
> The above table presents the performance drops when we remove the proteins appearing in downstream tasks from ProteinKG25. We see that the pre-trained model shows slightly worse results on three out of six downstream tasks: contact prediction, homology detection, and affinity prediction, while performing competitively on PPI, stability prediction, and semantic similarity inference. **These comparisons imply that we may be fine with the consequences of data leakage**. We will continue to investigate this problem in future work.
>
> ----
> ### Comment-3:
> > Related to semantic gaps: The authors claim that semantic gaps are the main issues of OntoProtein but it is very hard to see any evidence demonstrated in the paper to support that claim. Since this is the main claim against OntoProtein, this needs to be supported by a piece of strong evidence with careful analysis.
>
> ### Response-3:
> Thanks for your insightful advice. We agree that 'semantic gaps' is not a clear indication of the problem of OntoProtein. In the revision, we point out a potential drawback of OntoProtein: it fails to explore protein knowledge graphs at a more granular level. Specifically, OntoProtein adopts mean pooling to compute contextualized representations for protein (averaging amino acid representations) and textual knowledge (averaging word representations). However, KeAP directly applies cross-modal attention to representations of amino acids and natural language words, leading to protein and knowledge exploration at a more granular level.

---

> > ### Author Response · Authors · 2022-11-17
> > **Response to Reviewer mLUJ (PART 2)**
> >
> > ### Comment-4:
> > > Reproducity: I checked both the paper and the appendix version, this information is needed to clarify the implementation details: a) what is the representation of proteins used in the encoder? I understand it is a bert-like encoder but whether you use pretrained models or train everything from scratch? 2) potential leakage happens if proteins in the test sets of the benchmark datasets are not removed from the training, do you check the overlapping between test sets and the ProteinKG25?
> >
> > ### Response-4:
> >
> > We appreciate your efforts in helping improve the reproducity. Our answers are as follows:
> >
> > - We follow the pre-training protocol used in OntoProtein. We use ProtBert as the pre-trained model for our knowledge enhanced pre-training.
> >
> > - Previously, **we used the same pre-training data as used in OntoProtein [ICLR'22]. We also used the same data splits on contact prediction, homology detection, and stability prediction as used in OntoProtein.** According to your comments, we checked the overlap between the pre-training data (ProteinKG25) and downstream datasets. We found that there are 22,978 amino acid sequences that appear both in ProteinKG25 and all downstream datasets. Accordingly, 396,993 triplets are removed from ProteinKG25, and the size of the new dataset is 91.87% of the original. We report the experimental results in the updated manuscript based on the new pre-trained model trained with the filtered ProteinKG25. We also conduct ablation studies to investigate the impact of the leaked protein (please refer to section 5.1 for more details). We find that the pre-trained model shows slightly worse results on three out of six downstream tasks: contact prediction, homology detection, and affinity prediction, while performing competitively on PPI, stability prediction, and semantic similarity inference. **These comparisons imply that we may be fine with the consequences of data leakage**. We will continue to investigate this problem in future work.

---

> ### Author Response · Authors · 2022-11-18
> **Follow up**
>
> Dear Reviewer mLUJ,
>
> Thank you for your time and effort in reading our response! We hope our response has addressed your concerns. If you still feel unclear or concerned, please let us know, and we will be more than glad to clarify and discuss any further concerns. If you feel your concerns have been addressed, please kindly consider if it is possible to update your score.
>
> Thank you!
>
> Paper1625 Authors

---

> ### Comment · Reviewer_mLUJ · 2022-11-24
> **Some of the concerns have been addressed**
>
> Dear authors,
> I read your responses to other reviewers and also new experiments you have done with leakage removed and adding results of ESM-1b. It is interesting to see that leakage does happen and that influences the reported results. Happy to see new corrected results and more information on reproducibility.
>
> On the other hand, I think the paper is an incremental work compared to the OntoProtein paper with a minor change in the learning architecture for which I agree with the other reviewers. Even so, I am happy to raise my score to "marginally above the acceptance threshold" as the authors have addressed some technical concerns  plus considering the limited novelty of the work.
>
> All the best,

---

> > ### Author Response · Authors · 2022-11-24
> > **Reply to Reviewer mLUJ**
> >
> > Dear Reviewer mLUJ,
> >
> > It is really a pleasant experience to receive your positive response on Thanksgiving Day. Again, we sincerely appreciate your insightful comments, which have greatly helped us to improve our manuscript!
> >
> > Happy Thanksgiving!
> >
> > Best,
> >
> > Paper1625 Authors

---

### Official Review · Reviewer_6wQd · 2022-10-20

**Confidence:** 4
**Correctness:** 3
**Technical Novelty And Significance:** 2
**Empirical Novelty And Significance:** 2
**Recommendation:** 5

**Clarity, Quality, Novelty And Reproducibility:**

The paper often uses exaggerated and poorly-defined terms that over-emphasize the novelty of the method. These include  'implicit knowledge encoding', 'protein-centric representation', 'primary structure reasoning'  'proteincentric knowledge exploitation', etc. These are not established jargon in the field. Similarly, the complexity/significance of the paper's contribution is over-stated in section 3.2, which explains a standard attention mechanism. In modern ICLR papers, only a couple of sentences should be necessary to describe this. The details could be in an appendix.

The paper considers benchmarking tasks that are publicly available, so they are reproducible in that sense.

The paper modifies the architecture of a model that was presented previously at ICLR. This modification provides large performance improvements, but the change in architecture is standard (using a standard attention mechanism) and won't be of general interest to the ICLR community.


**Strength And Weaknesses:**

=Strengths=
Thorough evaluation on lots of downstream tasks
Lots of citations to recent work in NLP that uses similar ideas.
The general modeling idea (established in the OntoProtein paper) is good.

=Weaknesses=
Methodological contribution is very minor.
Writing is confusing. Lots of poorly-defined jargon (see below).
Potential issues regarding data leakage for test set (see below).


**Summary Of The Paper:**

LM pretraining on protein sequences has become popular in recent years. One difference between proteins and NLP is that for proteins there is often lots of structured/unstructured knowledge graph information for each pretraining example. This paper extends masked language modeling (BERT/MLM) by allow the model to also attend to an encoding of textual side information. This improves the performance of models on a variety of benchmark tasks.

**Summary Of The Review:**

Can you comment on the relationship between your work and ProGen (https://arxiv.org/abs/2004.03497)? This also uses protein KG information in a large language model.

When you use your model to get per-residue protein embeddings for downstream tasks, I'm assuming that you pass correct / ground truth KG information. For some of the tasks, does this present a data leakage problem, where there is information provided at evaluation time that reveals information about test-set labels? For example, for the protein-protein interaction task, you're providing GO terms for proteins in the test set. With this, are you effectively learning a model about which GO terms interact with other GO terms?

I was disappointed that, besides Table 7, the experiments don't provide a direct ablation that removes the ontology information. The ProteinBert results are from a separate paper. Surely there are other differences between the systems, such as hyper-parameters.

---

> ### Author Response · Authors · 2022-11-17
> **Responses to Reviewer 6wQd (PART 1)**
>
> We sincerely appreciate your time in reviewing our manuscript. Below please find our point-by-point response to address your concerns. Please do not hesitate to contact us if you have further questions or suggestions.
>
> -----
> ### Comment-1
> >Methodological contribution is very minor. The paper modifies the architecture of a model that was presented previously at ICLR. This modification provides large performance improvements, but the change in architecture is standard (using a standard attention mechanism) and won't be of general interest to the ICLR community.
>
> ### Response-1
> We are afraid that the reviewer has probably misunderstood the methodological contribution of our paper. There are two notable architectural modifications made by KeAP, which help achieve large improvements on a range of downstream tasks. We clarify them as follows:
>
> 1. A neat encoder-decoder architecture for knowledge-based protein pre-training. Specifically, KeAP replaces the contrastive learning module and the heuristic knowledge-aware negative sampling in OntoProtein with a multi-modal decoder architecture. As a result, the whole framework can be trained using the MLM objective only. Here, our methodological contribution is to show that a masked auto-encoder can be a competitive choice for protein pre-training on knowledge graphs.
>
> 2. Token-level exploration of textual side information (i.e., gene ontologies). OntoProtein applies mean pooling (i.e., global average pooling) to amino acids and tokenized words to obtain contextualized representations of proteins and gene ontologies. However, we argue that such a global average operation cannot fully explore the token-level information. To address this concern, KeAP directly applies cross-modal attention on top of amino acids and tokenized words.
>
> As for the interest of the ICLR community, KeAP provides a neat solution for knowledge-based pre-training, where multi-modal side data can be utilized to enhance single-modal representations. To our knowledge, we are the first to propose using cross-modal attention to improve single-modal representations with multi-modal data.
>
> -----
> ### Comment-2:
> >The paper often uses exaggerated and poorly-defined terms that over-emphasize the novelty of the method. These include 'implicit knowledge encoding', 'protein-centric representation', 'primary structure reasoning' 'proteincentric knowledge exploitation', etc. These are not established jargon in the field.
>
> ### Response-2:
> Thanks for your comments. We have removed terms like 'implicit knowledge encoding', 'protein-centric representation', and 'proteincentric knowledge exploitation' in the updated manuscript. We replace 'primary structure reasoning' with 'primary structure modeling' for clarification, where 'primary structure' refers to the amino acid sequence (cf. [wiki](https://en.wikipedia.org/wiki/Protein_primary_structure)).
>
> -----
> ### Comment-3:
> > Can you comment on the relationship between your work and ProGen (https://arxiv.org/abs/2004.03497)? This also uses protein KG information in a large language model.
>
> ### Response-3
>
> Thanks for your comment. We have added ProGen (https://arxiv.org/abs/2004.03497) to the related work. There are two main differences between our work and ProGen:
>
> - Motivation (Generative vs. Discriminative): ProGen aims to train a language model for protein generation. In contrast, our goal is to learn transferable protein representations for downstream discriminative tasks (e.g., structure and interaction predictions).
>
> - Implementation (Uni-directional vs. Bi-directional): ProGen trains a unidirectional transformer, where a casual mask is used to preclude attending to future tokens. In contrast, our KeAP employs a bi-directional training strategy, where input amino acids are randomly masked. Besides, our method differs from ProGen in the way of utilizing gene ontologies. ProGen prepends ontology terms to an amino acid sequence, while our KeAP adopts cross-attention modules to incorporate gene ontologies.

---

> > ### Author Response · Authors · 2022-11-17
> > **Responses to Reviewer 6wQd (PART 2)**
> >
> > ### Comment-4:
> > > When you use your model to get per-residue protein embeddings for downstream tasks, I'm assuming that you pass correct / ground truth KG information. For some of the tasks, does this present a data leakage problem, where there is information provided at evaluation time that reveals information about test-set labels? For example, for the protein-protein interaction task, you're providing GO terms for proteins in the test set. With this, are you effectively learning a model about which GO terms interact with other GO terms?
> >
> > ### Response-4:
> >
> > Thanks for your insightful comments. Previously, **we used the same pre-training data as used in OntoProtein [ICLR'22]. We also used the same data splits on contact prediction, homology detection, and stability prediction as used in OntoProtein.** During the rebuttal period, we checked the overlap between the pre-training data (ProteinKG25) and downstream datasets. We found that there are 22,978 amino acid sequences that appear both in ProteinKG25 and all downstream datasets. Accordingly, 396,993 triplets are removed from ProteinKG25, and the size of the new dataset is 91.87% of the original.
> >
> > We report the experimental results in the updated manuscript based on the new pre-trained model trained with the filtered ProteinKG25. We also conduct ablation studies to investigate the impact of the leaked protein (please refer to section 5.1 for more details). We provide an excerpt here for your consideration:
> >
> > ||Contact|Homology|Stability|PPI|Affinity|Semantic Similarity|
> > |----|----|----|----|----|----|----|
> > |Performance changes|-1%|-1%|0%|+1|-1%|0%|
> >
> > The table above presents the performance drops when we remove the proteins that appear in downstream tasks from ProteinKG25. We see that the pre-trained model shows slightly worse results on three out of six downstream tasks: contact prediction, homology detection, and affinity prediction, while performing competitively on PPI, stability prediction, and semantic similarity inference. **These comparisons imply that we may be okay with the consequences of data leakage**. We will continue to investigate this problem in future work.
> >
> > ----
> > ### Comment-5:
> > > I was disappointed that, besides Table 7, the experiments don't provide a direct ablation that removes the ontology information.
> >
> > ### Response-5:
> > Sorry for the confusion. We have conducted an ablation study that completely removes the ontology information, denoted as *- PiK* in Table 7, which means we only train an auto-encoder with the MLM objective. We observe a 6% performance drop on the contact prediction task.

---

> ### Author Response · Authors · 2022-11-18
> **Follow up**
>
> Dear Reviewer 6wQd,
>
> Thank you for your time and effort in reading our response! We hope our response has addressed your concerns. If you still feel unclear or concerned, please let us know, and we will be more than glad to clarify and discuss any further concerns. If you feel your concerns have been addressed, please kindly consider if it is possible to update your score.
>
> Thank you!
>
> Paper1625 Authors

---

> ### Author Response · Authors · 2022-11-28
> **Looking Forward to Your Reply!**
>
> Dear Reviewer 6wQd,
>
> Thanks a lot for your time and efforts in reviewing our paper. We have tried our best to address all mentioned concerns. We would appreciate it if you could take a look at our response. If there are any new questions, we can therefore reply in time.
>
> Best,
>
> Paper1625 Authors

---

### Official Review · Reviewer_oJan · 2022-10-24

**Confidence:** 3
**Correctness:** 3
**Technical Novelty And Significance:** 3
**Empirical Novelty And Significance:** 2
**Recommendation:** 6

**Clarity, Quality, Novelty And Reproducibility:**

This paper is well-written and well-motivated. The code is without running scripts.

**Strength And Weaknesses:**

Strength:

1.This paper proposes a new knowledge-enhanced method of Knowledge-exploited Autoencoder for protein representation learning.

2.Experimental results show that the proposed approach achieves better performance than baselines.

3.This paper also provides ablations and discussions of failure cases, which help readers understand the model.

Weakness:

1.It looks like the authors have made some small improvements on the basis of the existing method OntoProtein. Specifically, both of these two methods aim to incorporate external domain knowledge (KG) into protein representation learning. It is questionable whether KeAP actually provides a great contribution or innovation to the community.

2.The three colors in the legend of the Figure1 are not very distinguishable, so it takes a lot of effort to distinguish the corresponding method of each color. In addition, I think the positions of the figures and the tables should be moved to the corresponding part of the text to make it easier for readers to read. For example, Figure1,2,3 and Table1,2,6,7,8. Figure3 is too small to be seen clearly, affecting the reader's understanding of the experiment.

3.I think there are some problems with the author's writing logic. Since the concepts of ProteinKG25 and Gene Ontology are mentioned for the first time in Section4, if readers are not familiar with the settings of OntoProtein, they will not know what relations are described in the protein-related knowledge graph.

**Summary Of The Paper:**

To overcome the semantic gap between protein sequences and natural language, this paper proposes KeAP (Knowledge-exploited Auto-encoder for Protein) to perform knowledge enhanced protein representation learning. Specifically, it performs protein-centric knowledge exploitation via the attention mechanism, and uses the MLM objective to make it easier to optimize. The generalization ability of the learned protein representation are evaluated by fine-tuning the pre-trained model on a wide range of downstream applications. Ablations and failure analyses are provided to facilitate the understanding of the model.

**Summary Of The Review:**

The paper introduces how domain knowledge can be implicitly incorporated into protein representation learning. The authors use several popular ideas, such as the attention mechanism and MLM objective and show its effectiveness on multiple downstream tasks. Unfortunately, the author's writing logic is not clear enough, making the article not particularly smooth to read.

---

> ### Author Response · Authors · 2022-11-17
> **Response to Reviewer oJan**
>
> We sincerely thank the reviewer for helping us improve the paper's organization and figure illustration. Below please find our point-by-point response to your insightful comments. We have incorporated your advice in the revision, making our paper more friendly to readers.
>
> ------------------
> ### Comment-1:
> > It looks like the authors have made some small improvements on the basis of the existing method OntoProtein. Specifically, both of these two methods aim to incorporate external domain knowledge (KG) into protein representation learning. It is questionable whether KeAP actually provides a great contribution or innovation to the community.
>
> ### Response-1:
> Thanks for your kind comments. We would like to clarify that the technical improvements made by KeAP are non-trivial and have significant impacts, although they may be easy to implement using existing deep learning libraries (e.g., the implementation of QKV attention). Precisely, our modifications mainly lie in how to leverage knowledge graph embeddings to enrich protein representations. In the following, we list these modifications and their potential benefits.
>
> *Modifications*:
> 1. On the granularity of knowledge graph embeddings: Mean Pooling (OntoProtein) vs. Token-level Attention (KeAP). OntoProtein adopts mean pooling to compute contextualized representations for protein (averaging amino acid representations) and textual knowledge (averaging word representations). However, KeAP directly applies QKV attention to representations of amino acids and natural language words, leading to protein and knowledge exploration at a more granular level.
>
> 1. On the way to incorporate knowledge: Contrastive Learning (OntoProtein) vs. Cross-modal Attention (KeAP). OntoProtein applied contrastive learning with knowledge-aware negative sampling (one main contribution of OntoProtein) to protein and knowledge graph embeddings. In contrast, our KeAP applies cross-modal attention to protein and knowledge graph embeddings.
>
> *Benefits (compared to OntoProtein)*:
> 1. KeAP explores knowledge graphs at a more granular level. OntoProtein fails to explore the token-level information in relation and attribute terms (i.e., gene ontology) because of the mean pooling operation applied to proteins and gene ontologies. In contrast, KeAP treats protein and gene ontology terms as sequences of tokens and applies cross-attention to leverage token-level information. We believe this may be the key reason why KeAP outperforms OntoProtein by large margins in a range of downstream tasks.
>
> 2. KeAP provides a neat solution for knowledge enhanced protein pre-training. The encoder-decoder architecture in KeAP can be trained using the MLM objective only (both contrastive loss and MLM are used in OntoProtein), making the whole framework easy to optimize and implement. We believe this characteristic of KeAP increases its potential to be treated as a widely adopted baseline in knowledge-based protein pre-training.
>
> 3. The methodology of KeAP can possibly be extended to a more general topic, i.e., enhancing single-modal representations with multi-modal data. Since the encoder-decoder architecture in KeAP can be trained using the MLM objective only, this neat design can be easily transferred to other domains (such as natural language processing) with minimal effort.
>
> ----
> ### Comment-2:
> >The three colors in the legend of the Figure1 are not very distinguishable, so it takes a lot of effort to distinguish the corresponding method of each color. In addition, I think the positions of the figures and the tables should be moved to the corresponding part of the text to make it easier for readers to read. For example, Figure1,2,3 and Table1,2,6,7,8. Figure3 is too small to be seen clearly, affecting the reader's understanding of the experiment.
>
> ### Response-2:
> Thanks for your efforts in making these suggestions. We have revised the manuscript according to your advice.
>
> ----
> ### Comment-3:
> >I think there are some problems with the author's writing logic. Since the concepts of ProteinKG25 and Gene Ontology are mentioned for the first time in Section4, if readers are not familiar with the settings of OntoProtein, they will not know what relations are described in the protein-related knowledge graph.
>
> ### Response-3:
> Thanks for your kind advice. We have added a preliminary section (section 3.1) in the revision. Please let us know whether you feel our background descriptions are sufficient.
>
> ----
> ### Comment-4:
> >The code is without running scripts.
>
> ### Response-4:
> Thanks for your comment. Previously, we included a README file in the supplementary material, where a template running script is provided with detailed illustrations of each input argument. To perform pre-training, you only need to run the following script:
> ```python
> sh script/run_pretrain.sh
> ```

---

> ### Author Response · Authors · 2022-11-18
> **Follow up**
>
> Dear Reviewer oJan,
>
> Thank you for your time and effort in reading our response! We hope our response has addressed your concerns. If you still feel unclear or concerned, please let us know, and we will be more than glad to clarify and discuss any further concerns. If you feel your concerns have been addressed, please kindly consider if it is possible to update your score.
>
> Thank you!
>
> Paper1625 Authors

---

### Author Response · Authors · 2022-11-18
**Clarifications and Summary of Revisions**

We sincerely appreciate the reviewers' efforts in providing constructive reviews and insightful suggestions. We have incorporated their constructive feedback as revisions to our paper. In the following, we clarify our technical contributions and summarize the changes made in the revised manuscript.

----
### Technical Novelty [oJan, 6wQd]

Compared to OntoProtein [ICLR'22], we have two technical contributions:

1. **We propose to explore the knowledge graph at a more granular level, i.e., token-level protein-ontology exploration via cross-modal attention**. This differs from OntoProtein, which applies contrastive learning to globally averaged representations of proteins and associated gene ontologies.

2. **We provide a neat solution not only to knowledge enhanced protein representation learning but also to the broad ICLR community, where multi-modal side data can be utilized to enhance single-modal representations**. To our knowledge, we are the first to propose using cross-modal attention to improve single-modal representations with multi-modal data.

----
### Concerns on Data Leakage [6wQd, mLUJ]

Previously, we used the same pre-training data (i.e., ProteinKG25) as used in OntoProtein [ICLR'22]. We also used the same data splits on contact prediction, homology detection, and stability prediction as used in OntoProtein.

After carefully checking the overlap between the ProteinKG25 and downstream datasets, we found that there are fewer than 10% of proteins that appear both in the pre-training and downstream data. Then, we retrain the model using the filtered pre-training data. We observed affordable performance drops after completely removing the overlapped protein from the pre-training data. The results on six downstream datasets are as follows:

||Contact|Homology|Stability|PPI|Affinity|Semantic Similarity|
|----|----|----|----|----|----|----|
|Performance changes|-1%|-1%|0%|+1|-1%|0%|

We see that the pre-trained model shows slightly worse results on three out of six downstream tasks: contact prediction, homology detection, and affinity prediction, while performing competitively on PPI, stability prediction, and semantic similarity inference. These comparisons imply that we may be okay with the consequences of data leakage. We will continue to investigate this problem in future work.


----
### Summary of Changes

**Experiment:**

- We reported the fine-tuning results of our KeAP in two settings. In the first setting, we pre-train the model using the full ProteinKG25 as in OntoProtein. In the second setting, we pre-train the model using the filtered ProteinKG25. The complete and filter training data can be accessed via [Google Drive](https://drive.google.com/file/d/1KHaf-FNXLSkm4KLPQSw5CwRA4TRF5Lsd/view?usp=sharing).

- We added the experimental results of ESM-1b. Compared to ESM-1b, our KeAP still achieves noticeable improvements in a range of downstream tasks, including contact prediction, homology detection, stability prediction, protein-protein interaction identification, and semantic similarity inference.

- We added ablation studies to demonstrate the performance differences between pre-trained models in two settings.

**Organization:**

We have updated the illustration figures, added a preliminary section, moved figures and tables to appropriate locations, and fixed minor typos.

We have revised some claims to make them more intuitive and reliable, following the advice from reviewer mLUJ.

---

### Decision · Program_Chairs · 2023-01-20

**Decision:**

Accept: poster

**Justification For Why Not Higher Score:**

We agreed with reviewers that the contributions are limited in scope and focus on a specific application. While protein successful representation learning methods can have a huge impact in real-world applications, this paper is not a breakthrough of the field.

**Justification For Why Not Lower Score:**

I believe the paper is worth a publication, but I understand if this paper makes space for other papers once the acceptance threshold is being drawn.

**Metareview: Summary, Strengths And Weaknesses:**

The paper addresses the problem of learning representations for proteins for a number of biological downstream tasks while exploiting background knowledge in the form of a knowledge graph. The proposed contribution extends a previous pipeline for this problem, OntoProtein, by proposing a number of engineering modifications -- a simplified loss and the use of cross-attention --  that proves to be successful in boosting the performances of several downstream tasks.

Reviewers appreciated the results and the efforts of the authors and overall liked the contribution. Some concerns about the presentation of paper (e.g., use of jargon, contributions can be summarized more) and its incrementality w.r.t. OntoProtein were raised. Furthermore, risks of potential data leakage were highlighted. During the rebuttal authors were quick to respond and addressed all the major concerns, including the leakage for which new experiments were run.

The only point of concern left was the incrementality of the work. I personally value the proposed engineering contributions as they, while seeming simple, require effort to be properly implemented and experiments rigorously carried out. I believe the work can be useful for people working on protein representations and it is worth presenting at the conference.

The paper is therefore accepted conditioned on the fact that the authors will include the discussion and results coming from the rebuttal phase. Furthermore they will address the jargon used in the paper as to simplify presentation.

**Note From Pc:**

if the above contains the word "oral" or "spotlight" please see: "oral" presentation means -> notable-top-5% and "spotlight" means -> notable-top-25%. As stated in our emails, we are disassociating presentation type from AC recommendations

**Summary Of Ac-Reviewer Meeting:**

The reviewers agreed that the paper contribution is novel but were questioning it to be incremental. This was the major point of discussion in the discussion (both via email and via zoom). Finally, the reviewers agreed that the engineering contributions and the experiments are not trivial, but their scope in a publication at a top-tier ML conference was still being questioned.

I personally value the proposed engineering contributions as they, while seeming simple, require effort to be properly implemented and experiments rigorously carried out. Furthermore, I believe that the leakage fixed in the discussion will be of interest for the next iteration of experiments that people in the community will focus on.